DOI: 10.1038/s41467-018-03167-2　　**OPEN**

# Attosecond recorder of the polarization state of light

Álvaro Jiménez-Galán[1], Gopal Dixit[2], Serguei Patchkovskii[1], Olga Smirnova [1,3], Felipe Morales[1] & Misha Ivanov[1,4,5]

High harmonic generation in multi-color laser fields opens the opportunity of generating isolated attosecond pulses with high ellipticity. Such pulses hold the potential for time-resolving chiral electronic, magnetization, and spin dynamics at their natural timescale. However, this potential cannot be realized without characterizing the exact polarization state of light on the attosecond timescale. Here we propose and numerically demonstrate a complete solution of this problem. Our solution exploits the extrinsic two-dimensional chirality induced in an atom interacting with the chiral attosecond pulse and a linearly polarized infrared probe. The resulting asymmetry in the photoelectron spectra allows to reconstruct the complete polarization state of the attosecond pulse, including its possible time dependence. The challenging problem of distinguishing circularly polarized, partially polarized, or unpolarized pulses in the extreme ultraviolet range is also resolved. We expect this approach to become the core ingredient for attosecond measurements of chiral-sensitive processes in gas and condensed phase.

[1] Max-Born-Institute, Max-Born Straße 2A, D-12489 Berlin, Germany. [2] Department of Physics, Indian Institute of Technology Bombay, Powai, Mumbai 400076, India. [3] Technische Universität Berlin, Ernst-Ruska-Gebäude, Hardenbergstraße 36A, 10623 Berlin, Germany. [4] Department of Physics, Humboldt University, Newtonstraße 15, D-12489 Berlin, Germany. [5] Department of Physics, Imperial College London, South Kensington Campus, SW7 2AZ London, UK. Correspondence and requests for materials should be addressed to Á.J.-G. (email: jimenez@mbi-berlin.de)

solated attosecond pulses give access to time-resolved studies of electronic response in atoms, molecules, surfaces, and transparent solids[1–6]. The attosecond streak camera[1,7–9] is the principal tool in such studies, thanks to its ability to both measure the attosecond response of matter and record linearly polarized pulses inducing this response. However, detecting more complex polarization states, for example, elliptic or partially polarized, with attosecond resolution, stands out as a major challenge, exacerbated by the pulses' ultrabroad bandwidth and the extreme ultraviolet (XUV)/soft-X-ray frequency range.

Solving this problem is both fundamentally interesting and timely. Circularly polarized high harmonics in the XUV and soft-X-ray range are now robustly produced[10–14], opening ways to generate isolated attosecond pulses with high ellipticity[14,15]. Given the coherent nature of high harmonic emission, the harmonics are expected to be perfectly polarized. Therefore, the recent measurements[16] were quite surprising: the analysis of elliptically polarized individual high harmonics produced by an ensemble of aligned nitrogen molecules suggested that these harmonics were only partially polarized (for example, due to fluctuations of the polarization ellipse). While for harmonics the result of Veyrinas et al.[16] is surprising, for isolated chiral attosecond pulses the emergence of partial polarization is less unexpected. Imperfect polarization of the attosecond pulse may arise, for example, from fluctuating carrier-envelope phases (CEP) of the pulses driving its generation in the bi-circular scheme[11,15]. Indeed, such CEP fluctuations lead to fluctuations in the polarization state of the generating two-color pulse, which would be mapped on the generated attosecond pulse. This makes unraveling complex polarization states of attosecond pulses without a priori assumptions particularly acute. Without such capability, the dream of attosecond studies of chiral-sensitive light–matter interaction, from electron dynamics in chiral molecules[17,18] to ultrafast magnetization and spin transport[19,20], cannot be realized. The challenge goes beyond pulses produced via high harmonic generation, encompassing free-electron laser (FEL)-based sources of few-femtosecond and, in the future, sub-femtosecond X-ray pulses.

Measuring the ellipticity and the polarization state of ultrashort XUV and soft X-ray pulses often relies on using mirrors with polarization-sensitive reflectivity, following in the steps of Etienne-Louis Malus dating back to 1809. However, these methods fail to distinguish, for example, between unpolarized and circularly polarized light[12], and lack temporal resolution[21]. For individual high-order harmonics, the degree of circularity has been measured using X-ray magnetic circular dichroism[11]. A complex train of attosecond pulses with rotating linear polarization has been reconstructed in the pioneering work[22] using photoemission from a solid surface (assuming, however, perfectly polarized individual harmonics). These approaches, however, do not allow for complete characterization of amplitudes, phases, and helicities across the ultrabroad spectrum of an isolated attosecond pulse with arbitrary carrier frequency and not only time-dependent, but also possibly partial polarization (for example, randomly fluctuating polarization ellipse).

In the following, we solve this problem by taking advantage of the extrinsic two-dimensional chirality induced in photo-absorption of the elliptically polarized XUV light in the presence of a linearly polarized infrared field (see Fig. 1). Intuitively, the polarization vector of the infrared (IR) field and the momentum vector of the photoelectron work like two hands of a clock, recording the rotation of the attosecond XUV pulse between them. The direction of rotation is reflected in the angle-resolved photoelectron asymmetry with respect to the IR polarization axis. This asymmetry relies on the attosecond timing of the XUV field rotation relative to the oscillations of the IR field.

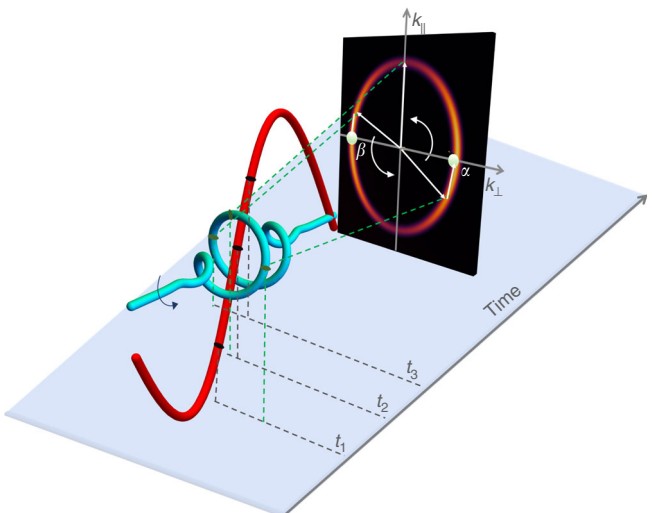

**Fig. 1** Semi-classical sketch of the origin of the photoelectron asymmetry. The IR pulse is polarized along the $k_\parallel$ direction and the photoelectron asymmetry is measured along the $k_\perp$ axis. The XUV pulse rotates counter-clockwise, the momentum distributions are observed at points α and β (green dots). The XUV-IR delay is set so that the IR vector-potential $\mathbf{A}_{IR}(t)$, which streaks the electron, changes its sign when the attosecond XUV pulse releases the electron in the direction of positive $k_\parallel$, at time $t_2$. At α, the path that contributes is that in which the photoelectron is ionized at time $t_1$ and streaked in the positive $k_\parallel$ direction. Half XUV cycle later, when the photoelectrons are released in the negative $k_\perp$ direction, the path that contributes to β is that in which the electron is ionized at time $t_3$ and streaked in the negative $k_\parallel$ direction. The asymmetry between the distributions at α and β is sensitive to the degree of coherence and relative phase between the two orthogonal components

Thus, it is only sensitive to the polarized component of the XUV. Combined with the linear measurement, for example of photo-ionization by the XUV pulse without the IR field, both the polarized and the unpolarized components can be retrieved. Below we show that, first, the photoelectrons are emitted asymmetrically with respect to the IR polarization direction. Second, this asymmetry encodes the XUV pulse polarization state. Third, we derive a simple analytical expression for it, confirming the theory with the full solution of the time-dependent Schrödinger equation (TDSE). Fourth, we reconstruct the complex time-dependent polarization states of two unknown attosecond pulses, including the contribution of an unknown random polarization component.

## Results

**Theoretical method.** Let an elliptically polarized attosecond XUV pulse ionize a target in the presence of a linearly polarized IR pulse (see Fig. 1), $z$ being their common propagation direction. Without loss of generality, we choose the laboratory $x$-axis to coincide with the major axis of the (time-integrated) XUV pulse polarization ellipse obtained from the linear measurement of the angle-resolved photoelectron spectrum. We write the partially polarized isolated attosecond XUV pulse as

$$\mathbf{F}_{XUV}(t) = \mathrm{Re}\left\{ F_X \boldsymbol{\sigma}(t) f_{XUV}(t - t_0) e^{-i\Omega(t - t_0)} \right\}. \quad (1)$$

Here $F_X$ is the electric field amplitude of the attosecond pulse, $\Omega$ is its carrier frequency, $f_{XUV}(t - t_0)$ is the envelope centered at some $t_0$, $\boldsymbol{\sigma}(t) = \boldsymbol{\sigma}_{pol}(t) \cos \gamma + \boldsymbol{\sigma}_{un} \sin \gamma$ describes the time-dependent polarization state, $\sin^2 \gamma$ characterizes the unknown weight of the unpolarized component, and $\boldsymbol{\sigma}_{pol}(t)$ and $\boldsymbol{\sigma}_{un}$ stand

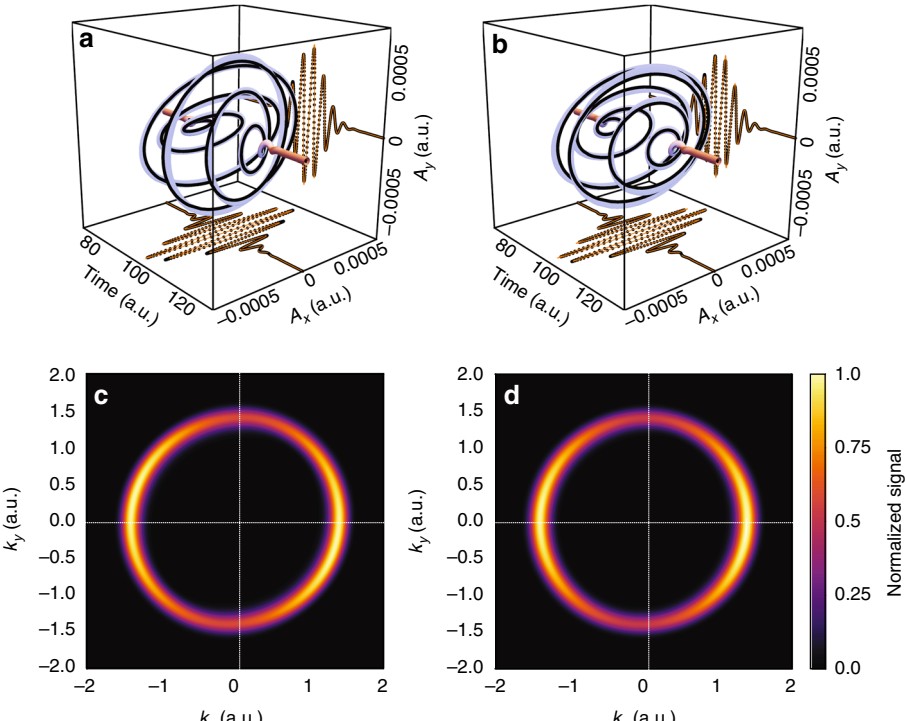

**Fig. 2** Reconstruction of two pulses with time-dependent polarization states. **a**, **b** Complete reconstruction of the time-dependent polarization state of two 250 as pulses carried at 41 eV (vector potential is shown). Ellipticity varies from $\varepsilon = 0.3$ along the $y$-axis to $\varepsilon = 0.625$ along the $x$-axis for the pulse in (**a**), and from $\varepsilon = 0.5$ to $\varepsilon = 0.9$ along the $y$-axis for the pulse in (**b**), see text for details. Original (purple in three dimensional (3D), orange in two dimensional (2D)) and reconstructed pulses (black lines in 3D, black dots in 2D). Additionally, the pulse in (**b**) has a 15% admixture of unpolarized light (not shown). **c**, **d** Normalized angularly resolved one-photon spectrum generated in a hydrogen atom using the pulse (**a**) for panel (**c**) and pulse (**b**) for panel (**d**)

for the polarized and unpolarized components,

$$\boldsymbol{\sigma}_{\text{pol}}(t) = \frac{\hat{\mathbf{e}}_x + i\varepsilon(t)\hat{\mathbf{e}}_y}{\sqrt{1 + \varepsilon(t)^2}} \ , \boldsymbol{\sigma}_{\text{un}} = \frac{1}{\sqrt{2}}\left[\hat{\mathbf{e}}_x + i\,e^{i\phi}\hat{\mathbf{e}}_y\right]e^{i\beta}, \quad (2)$$

where $\epsilon(t)$ is the time-dependent ellipticity of the field (see supplementary note 1). Random phases $\phi$ and $\beta$ in $\boldsymbol{\sigma}_{\text{un}}$ ensure that there is no correlation between $\boldsymbol{\sigma}_{\text{pol}}(t)$ and $\boldsymbol{\sigma}_{\text{un}}$ and between the $x$, $y$ components of $\boldsymbol{\sigma}_{\text{un}}$. One physical scenario leading to the presence of an unpolarized component could be the fluctuations in the relative phase or amplitude of $x$ and $y$ components of the attosecond pulse from shot to shot. By aligning the $x$-axis and $y$-axis of the laboratory frame with the major and minor axis, respectively, of the polarization ellipse obtained in the one-photon measurement, we ensure that the $x$ and $y$ components of the polarized part are $\pi/2$ out of phase (see supplementary note 5 for the discussion of a time-dependent phase between the $x$ and $y$ components). Here we assume that both polarized and unpolarized components have the same temporal envelope. In principle, one could also consider a more general case where these envelopes are not identical, leading to a time-dependent degree of depolarization.

Figure 2c, d shows the one-photon spectra generated in hydrogen by the attosecond pulses in Fig. 2a, b, respectively. Clearly, these spectra cannot be used to reconstruct the complex time-dependent polarization states of the input pulses (Fig. 2a, b), but they do yield the time-averaged orientation of the polarization ellipse.

We now add a linear IR field, polarized first along the major axis $\hat{\mathbf{e}}_x$ and then along the minor axis $\hat{\mathbf{e}}_y$ of the time-averaged ellipse. In each case, the photoelectron signals are recorded as a function of $\mathbf{k} = (k_{\parallel}, k_{\perp})$ in the XUV polarization plane, with $\parallel$

and $\perp$ meaning parallel and perpendicular to the IR polarization. Full angular resolution is not required: signals parallel and orthogonal to the IR polarization will be sufficient. Experimentally, nonetheless, angle-resolved measurements of attosecond photoionization in the presence of an IR field are now feasible[23]. The required X-ray / XUV-IR/THz pump-probe technology is also available for FEL-based sources[24], where complete characterization of FEL pulses of arbitrary polarization state is equally pertinent, see[25] for characterization of femtosecond FEL pulses.

First, we record the two standard FROG-CRAB (frequency-resolved optical gating for complete reconstruction of attosecond bursts) spectrograms[26] as function of the IR-XUV delay $t_0$, detecting the photoelectrons parallel to the IR polarization, for the two IR polarizations. These spectrograms enable complete reconstruction of the XUV field components $F_x(t)$ and $F_y(t)$ individually. Second, we need to characterize their mutual coherence. For this purpose, we use the photoelectron spectra recorded orthogonal to the IR polarization. The observables are the average signal $S$ and the asymmetry $A$,

$$S\big(k_{\parallel},k_{\perp},t_0\big) = \left(\big|a(k_{\parallel},k_{\perp},t_0)\big|^2 + \big|a(k_{\parallel},-k_{\perp},t_0)\big|^2\right)/2,$$

$$A\big(k_{\parallel},k_{\perp},t_0\big) = \frac{\big|a\big(k_{\parallel},k_{\perp},t_0\big)\big|^2 - \big|a\big(k_{\parallel},-k_{\perp},t_0\big)\big|^2}{S\big(k_{\parallel},k_{\perp},t_0\big)}, \quad (3)$$

where $\big|a(k_{\parallel},k_{\perp},t_0)\big|^2$ is the photoionization probability recorded for the XUV-IR delay $t_0$, and we will use signals for $k_{\parallel}{=}0$, minimizing the distortions caused by the Coulomb-laser coupling[27].

For ionization from an atomic $s$-orbital (for example, hydrogen or helium) and for the IR polarized along $\hat{\mathbf{e}}_x$, the asymmetry

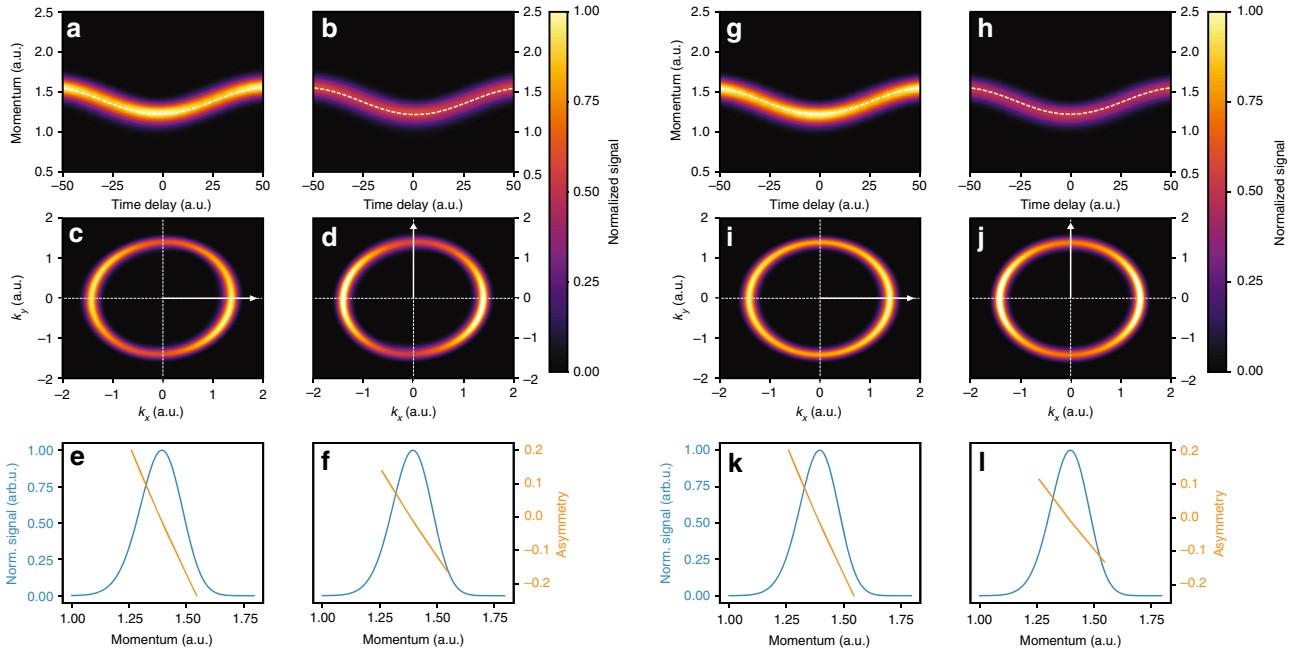

**Fig. 3** IR-streaked photoionization observables used in the reconstruction. Numerical spectra and observables for the laser-assisted photoionization of the hydrogen atom by the pulse in Fig. 2a (**a-f**) and by the pulse in Fig. 2b (**g-l**). **a**, **b**, **g**, **h** Normalized streaking trace, measured along the IR polarization axis. **c**, **d**, **i**, **j** Angularly resolved IR-streaked signal of the photoelectron distribution at the maximum of the IR electric field. The white arrow indicates the polarization direction of the IR laser. **e**, **f**, **k**, **l** Photoelectron asymmetries (orange curves) and average signals (blue lines) extracted from the angularly resolved spectrum on top in the direction perpendicular to the laser polarization axis. The orientation of the IR polarization axis with respect to the major axis of the XUV ellipse is parallel in (**a**), (**c**), (**e**), (**g**), (**i**), and (**k**), and perpendicular in panels (**b**), (**d**), (**f**), (**h**), (**j**), and (**l**)

measured along $\hat{\mathbf{e}}_y$ is (see supplementary note 2 for details)

$$
\begin{aligned}
A_1(k_\perp, t_0) \simeq \; & -\frac{1}{S_1(k_\perp, t_0)} \frac{\partial[S_1(k_\perp, t_0)/\sigma_{cs}(k_0)]}{\partial E_k} \\
& \times \frac{2\,\sigma_{cs}(k_0)\,F_{IR}(t_0)\varepsilon(t_0)\cos^2\gamma}{k_0\left[\varepsilon(t_0)^2\cos^2\gamma + \frac{1+\varepsilon(t_0)^2}{2}\sin^2\gamma\right]},
\end{aligned}
\tag{4}
$$

where the average signal $S_1$ is measured along $\hat{\mathbf{e}}_y$, $\mathbf{k}_0 \equiv \mathbf{k} + \mathbf{A}_{IR}(t_0) = \mathbf{k}$ at the zeros of $\mathbf{A}_{IR}$ where the asymmetry is measured, $\sigma_{cs}(k_0)$ is proportional to the cross-section, and $\varepsilon(t_0)$ is the ellipticity at the peak of the XUV pulse. For the IR polarized along $\hat{\mathbf{e}}_y$, we find

$$
\begin{aligned}
A_2(k_\perp, t_0) \simeq \; & -\frac{1}{S_2(k_\perp, t_0)} \frac{\partial[S_2(k_\perp, t_0)/\sigma_{cs}(k_0)]}{\partial E_k} \\
& \times \frac{2\,\sigma_{cs}(k_0)\,F_{IR}(t_0)\epsilon(t_0)\cos^2\gamma}{k_0\left[\cos^2\gamma + \frac{1+\epsilon(t_0)^2}{2}\sin^2\gamma\right]},
\end{aligned}
\tag{5}
$$

where the average signal $S_2$ is now measured along $\hat{\mathbf{e}}_x$, and $k_\perp$ is now along $\hat{\mathbf{e}}_x$. Note that $F_{IR}(t_0)$ is known from the first FROG-CRAB measurement. Now, the coherence $\cos^2\gamma$ between the two components and $\epsilon(t_0)$ are obtained by fitting the measured $A_1$ and $A_2$ to Eqs. (4, 5). Since the amplitudes and phases for $x$ and $y$ components separately are reconstructed from the standard streak-camera measurement, recovering the relative coherence of the two components at the peak of the XUV pulse gives complete information for the pulse as a whole, including time-dependent $\epsilon(t)$ (see Fig. 2a, b). Our approach thus requires the same CEP stability as a standard FROG-CRAB experiment.

**Reconstruction of attosecond pulses**. Figures 2 and 3 illustrate the method using ab initio numerical experiments for a hydrogen atom. The input pulses are shown in Fig. 2a, b and have complex

time-dependent polarization state. Moreover, while the first pulse is perfectly polarized ($\cos^2\gamma = 1$), the second pulse has unpolarized component ($\cos^2\gamma = 0.85$). The IR pulse carried at 800 nm was 7.2 fs long, with intensity $I_{IR} = 3.5 \times 10^{12}$ W/cm². To compute the photoelectron spectrum, we solved the TDSE numerically exactly (see Methods). Full reconstruction of the polarized parts of the pulses is shown as black lines and circles in Fig. 2a, b, demonstrating high accuracy even in the presence of an unpolarized XUV component.

Figure 3a–f shows the observables measured for the pulse in Fig. 2a, while those for the pulse in Fig. 2b are shown in Fig. 3g–l. The top panels in Fig. 3 show the standard streaking spectrograms measured along the IR polarization, for the IR-oriented parallel (a, c) and perpendicular (b, d) to the major axis of the XUV ellipse. From the spectrogram, we extract the intensity of the IR field $I_{IR}$ and the relative peaks of the XUV $x$ and $y$ spectrograms. For the pulse in Fig. 2a, we obtain $I_{IR} = 3.3 \times 10^{12}$ W/cm² and the peak of the $y$ component is retarded with respect to the $x$ component by $t_{0,y} - t_{0,x} = 70.7$ as. For the pulse in Fig. 2b, we find $I_{IR} = 3.3 \times 10^{12}$ W/cm² and $t_{0,y} - t_{0,x} = 22.2$ as. Using a standard FROG-CRAB retrieval algorithm[26], we may fully characterize these temporal profiles, including their duration and chirp. In the absence of chirp, one can also use the average signals recorded perpendicular to the IR polarization, minimizing distortions associated with the Coulomb-laser coupling[27] and finite-IR-pulse duration effects. These average signals are shown as the blue curves in the bottom panels of Fig. 3. For the pulse in Fig. 2a, they yield the durations of $\tau_x = 239$ as for the $x$ component and $\tau_y = 225$ as for the $y$ component. For the pulse in Fig. 2b, the extracted values were $\tau_x = 247$ as and $\tau_y = 243$ as. The different durations and time centers between the temporal profiles of the $x$ and $y$ components naturally make the ellipticity of the XUV pulse time dependent.

The central panels in Fig. 3 show the angle-resolved photoelectron spectra for the IR-XUV time delay where the asymmetry is strongest, that is, when the XUV pulse is centered at the peak of the oscillation of the IR laser electric field. To relate to a laboratory experiment, we integrate the signal over an 11 degree acceptance angle to extract the asymmetries $A_1$ and $A_2$ measured orthogonal to the IR polarization axis. The orange curves in the bottom panels of Fig. 3 show the asymmetries extracted from the spectrum. Even for the moderate IR intensity of $I_{IR} = 3.5 \times 10^{12}$ W/cm$^2$, the values are roughly 10%, well within experimental detection[28].

To obtain $\epsilon(t_0)$ and $\gamma$, we fit the asymmetries in Fig. 3 to the corresponding equations for $A_1$, $A_2$, extracting $\epsilon(t_0) = 0.784$ at the pulse center and $\cos^2 \gamma = 1$ for the pulse in Fig. 2a, and $\epsilon(t_0) = 0.691$ and $\cos^2 \gamma = 0.826$ for the pulse in Fig. 2b, in excellent agreement with the values used in our numerical simulation. We have checked that the method is robust against effects associated with CEP jitter, amplitude fluctuations between the $x$ and $y$ components, and a misalignment between the IR polarization axis and the axes of the time-averaged XUV polarization ellipse (see supplementary notes 4, 6, and Fig. 2c, where the misalignment is 15 degrees). We have also used different IR intensities, pulse durations, ellipticities, and degrees of polarization, obtaining equally good agreement (see supplementary note 3).

In conclusion, we have shown how to record the time-dependent polarization state of an isolated attosecond pulse along with its temporal structure, including the possibility to reconstruct perfectly polarized pulses with complex polarization state and distinguish them from partially polarized attosecond pulses. We have theoretically demonstrated both the validity and feasibility of our method. Extension of the method to the case of time-dependent relative phase between the $x$ and $y$ directions of the polarized component of the field is also possible, but the pulse reconstruction requires numerical fitting of TDSE simulations to the experimentally measured asymmetries (see supplementary note 5). Our work opens the way to monitor and control light–matter interactions of chiral nature with attosecond precision.

## Methods

**Ab initio numerical experiments**. The numerical results have been obtained by propagating the TDSE numerically for the hydrogen atom, using the code described in ref.[29]. We have used a radial box of 1971 a.u., with a total number of points $n_r = 5000$. We use a uniform grid, with 50 points (grid spacing of 0.096 a.u.) at the origin, followed by 71 points on a logarithmic grid, with a scaling parameter of 1.02, starting at 4.90 a.u., and finally 4879 points on a uniform grid with a spacing of 0.4 a.u. We placed a complex boundary absorber at the border of the radial box (starting at 1939 a.u.) in order to avoid reflections. However, the box is sufficiently large to contain the full wave function at the end of the pulse (we checked that the total norm in the simulation volume is 1.0 at the end of the pulse). Therefore, we can apply the iSURFC method[30]. The maximum angular momenta included in the spherical harmonics expansion was $\ell_{max} = 30$. The time grid had a spacing of $dt = 0.025$ a.u. All the discretization parameters have been checked for convergence.

The photoelectron signal generated by a partially polarized XUV pulse in the presence of a linear IR field was simulated as

$$I_{tot} = I_{pol}(\epsilon(t)) \cos^2 \gamma + \frac{I_\perp + I_\parallel}{2} \sin^2 \gamma, \qquad (6)$$

where $I_{pol}(\epsilon(t))$ is the photoelectron signal of a simulation with an elliptically polarized XUV pulse (with degree of polarization of one) plus a linearly polarized IR field, $I_\perp$ is the signal of a simulation with a linear XUV pulse plus a linear IR field polarized in the direction perpendicular to the XUV polarization axis, $I_\parallel$ is the signal of a simulation with a linear XUV pulse plus a linear IR field polarized along the direction of the XUV, and $\cos^2 \gamma$ is the degree of polarization.

**Data availability**. The data that support the findings of this study are available from the corresponding author upon request.

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

## Acknowledgements

A.J.-G. and M.I. acknowledge support from the DFG QUTIF grant IV 152/6-1. M.I. acknowledges support from the EPSRC/DSTL MURI grant EP/N018680/1. G.D. acknowledges the Ramanujan fellowship (SB/S2/ RJN-152/2015). O.S. acknowledges support from the DFG QUTIF grant SM 292/2-3.

## Author contributions

A.J.-G. and F.M. have carried out the calculations. M.I., A.J.-G., G.D. and O.S. have developed the ideas and formalism. S.P. has developed the TDSE code and provided initial calculations. A.J.-G. and M.I. wrote the main part of the manuscript, which all the authors discussed.

## Additional information

**Competing interests:** The authors declare no competing financial interests.

