## [Peer Review File · Nature Communications]

Reviewers' comments:

Reviewer #2 (Remarks to the Author):

I consider that the revised manuscript now submitted to Nature Communications has been greatly improved with respect to the originally submitted version, in terms of clarity and structure, as well as in the strength and applicability of the presented simulated results, and I am glad that the authors found the comments and criticism helpful for improving the quality of the work.

The detailed reply provided by the authors adequately addresses most of my original concerns. The distinction between the method they introduce regarding the measurement of isolated attosecond pulses with a broadband spectrum (versus methods that measure an attosecond pulse train with a discrete spectrum) is now clear, and previous inaccuracies, such as claiming that the method is single-shot, have been totally removed. The 3D schematics now given in Figs. 1 and 2 enable a quicker understanding of the underlying geometry and physics, which will be greatly appreciated by readers. They have considered potential issues in an actual experiment (detailed in the SI), namely the alignment of the IR field and the fluctuations of the polarization ellipse due to fluctuations of the CEP of the input pulses. Here, I would add that fluctuations can also arise due to timing jitter between the pulses in a real experiment involving multiple input pulses.

Although I agree that CEP stability as required by a FROG-CRAB measurement is sufficient for retrieving an isolated attosecond pulse with arbitrary polarization, the proposed method is not intrinsically single shot, and will inevitably record photoelectron spectra averaged over many such attosecond pulses that naturally present fluctuations between one another. Therefore, and in the lack of a proof-of-principle experiment, it is important to extend the analysis by calculating the effect of typical CEP-noise on the resulting averaged measurements and to quantify its effects on the precision and accuracy of the retrievals in the more realistic and representative case of an averaged measurement, which will consequently affect the results summarised in Table 1 of the SI.

Reviewer #4 (Remarks to the Author):

The authors present a methodology for the complete characterizations state of attosecond high harmonic pulses utilizing attosecond streaking.

First I would like to say that the topic is very timely as the idea to explore magnetic phenomena and dichroism in molecules is now coming online.

Second the technique presented is intuitive, relies on existing principles but with the appropriate critical modifications, which can make such a metrology possible. I would raise a number of concerns similar to these of the other two reviewers.

Yet I believe that the revised manuscript corrects the problems of the earlier version indeed. The manuscript relies and properly cites previous research properly. I believe that the technique will become an important addition in the toolbox of ultrafast x-ray science.

I would like to recommend the revised manuscript for publication in Nature communications.

Reviewer #5 (Remarks to the Author):

The authors present a novel approach for measuring the (time-dependent) polarization state of isolated attosecond pulses based on an extension of "standard" attosecond streaking techniques. The authors promise a "complete solution" to the problem of characterizing attosecond pulses with complex polarization state. Judging from the referee reports in the previous round, this statement was highly dubious, and even with the apparently significant improvements implemented in response to the reports, I am not convinced that this statement (and several similar ones in the main text) is actually justified.

In the following, I discuss some major and minor points that I think should be addressed:

1. Primarily, the attosecond pulse as written by the authors does not describe a completely arbitrary time-dependent polarization state, as far as I can tell. For one, the phase relation between x and y components of the polarized part is fixed to be fully out of phase (as the y-direction has a prefactor i , while $\epsilon(t)$ is real as far as I can tell). This means that the pulses described by the authors cannot represent a situation where, e.g., the pulse is always linearly polarized, but the polarization direction changes (as previously mentioned by one of the referees). This could be remedied by allowing ϵ to be complex, or equivalently introducing an additional factor $e^{i\psi(t)}$. With this extension, the authors could describe the "polarized" component of a pulse with arbitrary $F_x(t)$ and $F_y(t)$.

2. In addition, the authors only use a time-independent (de)polarization degree γ , such that they cannot describe, e.g., a situation where first an unpolarized prepulse arrives, followed by a perfectly polarized main pulse.

3. It is not fully clear to me whether the assumption that the "unpolarized" contribution has equal x and y components is necessary or physically justified. According to my understanding, this term describes a randomly fluctuating contribution to the pulses, but I could just as well imagine a situation where there is such a contribution, but it is always linearly polarized in the same direction. E.g., this could occur in "standard" HHG with linear polarization of the driving pulses (such that all generated pulses are also linearly polarized), but where the IR pulse CEP is not well-stabilized, such that each attosecond pulse is slightly different.

4. The authors claim that it is basically trivial to obtain the x- and y-components of the pulse separately ("These spectrograms enable complete reconstruction of the XUV field components $F_x(t)$ and $F_y(t)$ individually.") However, as far as I see, it is never mentioned what $F_x(t)$ and $F_y(t)$ actually are here. Since the unpolarized contribution has a random phase relation to the polarized part, the x-component and y-component of the full pulse also cannot be described by single functions $F_x(t)$ / $F_y(t)$, but themselves consist of mixtures of "coherent" and "incoherent" radiation with a random phase between them. If the supposedly reconstructed $F_x(t)$ and $F_y(t)$ only corresponds to the coherent part, having these two functions already fully describes the "polarized" part of the pulse, and ϵ (and even a potential additional phase ψ as mentioned above) could be calculated from these functions trivially. In that case, it is not clear to me why an additional measurement would still be necessary. On the other hand, if the XUV field components mentioned in the text correspond to some mixture between the coherent and incoherent parts, it should be explained what is actually measured.

If the authors could show that their approach can address all of these points and they can still reconstruct the full pulse, I believe their very strong statements about a "complete solution" would indeed be justified. Otherwise, I would recommend that the limitations are very clearly stated and the

language is toned down a bit, even though I still believe that the method is interesting and could be useful.

In the following, I mention some more minor points, mostly related to unclear descriptions that could be improved in my opinion.

5. The isolated attosecond pulses are described by complex functions, and it is not mentioned that the real part should be taken to obtain the actual electric field (which must be real).

6. The authors talk of "the" isolated attosecond pulse, but then use a description that is clearly meant to be understood statistically, i.e., as describing a large ensemble of similar pulses, where the "polarized" part is identical each time, while the "unpolarized" part has a random phase in each realization. However, this is never explicitly mentioned (although it is acknowledged in their reply to the previous referee reports). Taking their statements as written, "the attosecond pulse" actually has a perfectly well-defined polarization state that simply depends on ϕ and β . It is only under the assumption that there exist many copies of the pulse with different random phases ϕ and β that one could talk of a "polarized" and "unpolarized" contribution. This should be made very clear, as it clearly also has led to some confusion in the discussion with the referees.

7. Since the "unpolarized" part has a random phase $e^{i\phi}$ between x and y components, it seems superfluous to include an additional prefactor "i" before the exponential (as this can be removed by shifting ϕ by $\pi/2$). The same applies to the polarized part, should the authors follow my suggestion above and extend the method to allow arbitrary phase relations between the x and y components there as well.

To summarize, I think the manuscript presents an interesting approach, but still has too many flaws to recommend publication in Nature Communications. If the limitations discussed above (and partially already pointed out by previous referees) cannot be lifted, I am not convinced it passes the barrier of importance to warrant publication in Nature Communications.

Answer to Referee 2

I consider that the revised manuscript now submitted to Nature Communications has been greatly improved with respect to the originally submitted version, in terms of clarity and structure, as well as in the strength and applicability of the presented simulated results, and I am glad that the authors found the comments and criticism helpful for improving the quality of the work. The detailed reply provided by the authors adequately addresses most of my original concerns. The distinction between the method they introduce regarding the measurement of isolated attosecond pulses with a broadband spectrum (versus methods that measure an attosecond pulse train with a discrete spectrum) is now clear, and previous inaccuracies, such as claiming that the method is single-shot, have been totally removed. The 3D schematics now given in Figs. 1 and 2 enable a quicker understanding of the underlying geometry and physics, which will be greatly appreciated by readers. They have considered potential issues in an actual experiment (detailed in the SI), namely the alignment of the IR field and the fluctuations of the polarization ellipse due to fluctuations of the CEP of the input pulses.

We thank the referee for his/her consideration of our revised work and positive feedback.

1. *Although I agree that CEP stability as required by a FROG-CRAB measurement is sufficient for retrieving an isolated attosecond pulse with arbitrary polarization, the proposed method is not intrinsically single shot, and will inevitably record photoelectron spectra averaged over many such attosecond pulses that naturally present fluctuations between one another. Therefore, and in the lack of a proof-of-principle experiment, it is important to extend the analysis by calculating the effect of typical CEP-noise on the resulting averaged measurements and to quantify its effects on the precision and accuracy of the retrievals in the more realistic and representative case of an averaged measurement, which will consequently affect the results summarised in Table 1 of the SI.*

We agree with the Referee. We have tested the robustness of our technique by considering the effect of typical CEP-noise on the resulting averaged measurements. Specifically, we have performed the reconstruction for the same pulse as shown in Fig.2b of the manuscript, but now taking into account CEP jitter of 0.1 rad, typical in attosecond pump-probe experiments. To this end, we have performed five calculations for different CEP of the IR field: -0.1, -0.05, 0, 0.05 and 0.1 rad, and summed the photo-electron spectra incoherently, thus mimicking the experimental jitter. We have applied the same procedure as for the case of zero jitter, i.e. we have used the asymmetry in the spectra, now extracted from the jitter-averaged spectra. The results are shown in Fig. 1, together with the fit to our analytical formula, which is indistinguishable from the simulations. The extracted parameters are summarized in Table 1, demonstrating robustness of our method. We have now included this study in the Supplementary Material.

Figure 1: Asymmetries A_1 and A_2 (as defined in the manuscript) for the pulse in Fig.2b of the manuscript. The asymmetries are obtained for the case when there is an admixture of 15% of de-polarized light. The spectra are averaged over 0.1 rad CEP jitter. The extracted values of the ellipticity and degree of polarization are given in Table 1 and compared to the case of perfect CEP stability.

	Theoretical	Reconstructed (no jitter)	Reconstructed (jitter)
$\epsilon(t_0)$	0.7	0.691	0.689
$\cos^2 \gamma$	0.85	0.826	0.826
$\epsilon(t_0)$	0.7	0.686	0.682
$\cos^2 \gamma$	1	0.99	0.99

Table 1: Comparison between the input $\epsilon(t_0)$ and $\cos^2 \gamma$ values (left column) and the reconstructed values assuming perfect CEP stability (central column) and with the CEP jitter of 0.1 rad (right column). The pulse considered has time-dependent polarization as in Fig.2b of the main manuscript.

We note that one strong point of this method is its redundancy. The asymmetry depends on both the XUV-IR time delay and the angle; and the analytical formula can be easily derived for arbitrary time delay and arbitrary angle. It is only for a matter of clarity that we have presented the reconstruction using one time delay and along one specific direction. If the experiment is fully angularly resolved (e.g., when using a COLTRIMS apparatus), one can extract the asymmetry at many XUV-IR time delays and along many angles. This will further reduce the impact of experimental uncertainties such as the CEP jitter.

Answer to Referee 4

The authors present a methodology for the complete characterizations state of attosecond high harmonic pulses utilizing attosecond streaking. First I would like to say that the topic is very timely as the idea to explore magnetic phenomena and dichroism in molecules is now coming online. Second the technique presented is intuitive, relies on existing principles but with the appropriate critical modifications, which can make such a metrology possible. I would raise a number of concerns similar to these of the other two reviewers. Yet I believe that the revise manuscript corrects the problems of the earlier version indeed. The manuscript relies and properly cites previous research properly. I believe that the technique will become an important addition in the toolbox of ultrafast x-ray science. I would like to recommend the revised manuscript for publication in Nature communications.

We would like to thank the referee for the consideration of our revised work and for acknowledging the importance of our technique as a tool for future ultrafast science.

Answer to Referee 5

The authors present a novel approach for measuring the (time-dependent) polarization state of isolated attosecond pulses based on an extension of "standard" attosecond streaking techniques. The authors promise a "complete solution" to the problem of characterizing attosecond pulses with complex polarization state. Judging from the referee reports in the previous round, this statement was highly dubious, and even with the apparently significant improvements implemented in response to the reports, I am not convinced that this statement (and several similar ones in the main text) is actually justified. In the following, I discuss some major and minor points that I think should be addressed:

If the authors could show that their approach can address all of these points and they can still reconstruct the full pulse, I believe their very strong statements about a "complete solution" would indeed be justified. Otherwise, I would recommend that the limitations are very clearly stated and the language is toned down a bit, even though I still believe that the method is interesting and could be useful. To summarize, I think the manuscript presents an interesting approach, but still has too many flaws to recommend publication in Nature Communications. If the limitations discussed above (and partially already pointed out by previous referees) cannot be lifted, I am not convinced it passes the barrier of importance to warrant publication in Nature Communications.

We thank the referee for his/her insightful and critical review of our work. We have performed the requested analysis. We have addressed the key concerns by extending our method to treat the case of arbitrary time-dependent phase between the x and y components of the field, demonstrating that we are also able to reconstruct the pulse in this scenario. With these new results, we believe we have lifted the limitations to which the referee refers to. This completes the analysis of virtually all realistic scenarios of complex attosecond pulse shapes, except for the case when the de-polarized component changes dramatically on the attosecond time-scale. Such case is, of course, extremely unlikely. Below, we address the referee's remarks point by point.

1. *Primarily, the attosecond pulse as written by the authors does not describe a completely arbitrary time-dependent polarization state, as far as I can tell. For one, the phase relation between x and y components of the polarized part is fixed to be fully out of phase (as the y -direction has a prefactor i , while $\epsilon(t)$ is real as far as I can tell). This means that the pulses described by the authors cannot represent a situation where, e.g., the pulse is always linearly polarized, but the polarization direction changes (as previously mentioned by one of the referees). This could be remedied by allowing ϵ to be complex, or equivalently introducing an additional factor $e(i\psi(t))$. With this extension, the authors could describe the "polarized" component of a pulse with arbitrary $F_x(t)$ and $F_y(t)$.*

We thank the Referee for this very useful comment. We have been able to fully address this concern in the new version of the manuscript.

First, we stress that the manuscript did already consider highly complex time-dependent polarization states, where the pulse started elliptic in one direction and then rotated to become elliptic in the orthogonal direction. Nevertheless, we agree with the Referee that in our derivation we have assumed that x and y components of the polarized field are shifted by $\pi/2$. In the following, we show that our method also works for arbitrary phase between the x and y components, *even* when this phase is time-dependent. The results are added to the supplementary material, clarified on the manuscript, and presented below for the convenience of the Referee.

Time-independent phase

Consider first the simple case of arbitrary, but time-independent phase shift between the x and y components:

$$\begin{aligned} F_x(t) &\propto \cos[\omega_{xUV}(t - t_0)], \\ F_y(t) &\propto \epsilon \sin[\omega_{xUV}(t - t_0) + \varphi_0]. \end{aligned} \quad (1)$$

From the linear measurement, we can find the major and minor axes of the time-averaged polarization ellipse. They will be rotated by the angle α relative to the x and y axes:

$$\tan 2\alpha = \frac{2\epsilon \sin \varphi_0}{1 - \epsilon^2}. \quad (2)$$

where ϵ is the ratio of the x and y components in the original frame. It is easy to check that, once we rotate our axes by α , the field components along the rotated x' and y' axes will have the $\pi/2$ phase-shift. Now we can apply our technique as before and characterize the pulse in the rotated frame (x', y') . Of course, the true value of the ellipticity ϵ' now differs,

$$\epsilon' = \frac{\epsilon \cos \alpha \cos \varphi_0}{\epsilon \sin \varphi_0 \sin \alpha + \cos \alpha}, \quad (3)$$

while the phase ϕ is

$$\tan \phi = \frac{\epsilon \cos \varphi_0 \sin \alpha}{\epsilon \sin \varphi_0 \sin \alpha + \cos \alpha}. \quad (4)$$

Importantly, we will also already know the degree of de-polarization, provided by our technique.

As a test case with time-dependent polarization, we used the pulse defined by

$$\begin{aligned} F_x &= \frac{f_{xUV}(t - t_{0xUV})}{\sqrt{1 + \epsilon(t)^2}} \cos[\omega_{xUV}(t - t_{0xUV})], \\ F_y &= \frac{\epsilon(t) f_{xUV}(t - t_{0xUV})}{\sqrt{1 + \epsilon(t)^2}} \sin[\omega_{xUV}(t - t_{0xUV}) + \varphi_0], \end{aligned} \quad (5)$$

where $\epsilon(t) = 0.9 + 0.2 \operatorname{Erf}[0.2(t - t_{0xUV})]$ and φ_0 was randomly chosen between 0 and $\pi/2$, with the value $0.22(\pi/2)$.

First, we performed the linear measurement, see Fig. 1a. We clearly see the tilt in the polarization ellipse. Hence, the x and y components of the pulse are not shifted by $\pi/2$. We then rotated the reference frame so that the major axis in the linear measurement coincides with the x axis (and the minor axis coincides with the y axis). This is shown in Fig. 1b. The angle of rotation found from the linear measurement was $\alpha = 36.2^\circ$. It coincides with that predicted by Eq. 2. In this rotated frame the x' and y' components of the field are shifted by $\pi/2$, and we apply our method. To obtain the reconstructed pulse in the original frame, we just need to rotate it back by angle α . The comparison between the pulse in Eq. 5 and the reconstructed pulse in the original reference frame (x, y) is shown in Fig. 1c.

Figure 1: (a) Angularly-resolved XUV-only photoelectron spectra for the pulse in Eq. 5. (b) Same photoelectron distribution as in panel (a) but with an $\alpha = 36.2^\circ$ rotation of the axes. (c) Pulse in Eq. 5 (orange are 2D projections) and its reconstruction (black line in 3D and black dots in 2D). The reconstruction is shown in the reference frame of panel (a).

Time-dependent phase

Now let us consider the time-dependent phase, $\varphi(t)$. Without the loss of generality, we set the $\varphi(t_0) = 0$, since we already know that this can be accomplished by simply rotating the reference frame.

First, we note that time-dependence associated with the phase $\varphi(t)$ can be obtained from the standard FROG-CRAB spectrograms, measured separately for both x and y directions. This time dependence results in frequency shifts and chirps. It is only the degree of coherence and the degree of de-polarization between the two components that are missing in the FROG-CRAB measurement. In other words, in general there could still be a component, with the weight $\sin^2 \gamma$, in which $\varphi(t)$ is random from shot to shot. This de-polarized component of the light can have any polarization between x and y with equal probability. To obtain $\sin^2 \gamma$, we need an additional observable – this is the left-right asymmetry proposed by us. Below we show that it works.

Figure 2: Fit to the streaking spectrograms along the (a) $k = k_x$ axis and along the (b) $k = k_y$ axis. For (a), the fit yields $\omega_{x,XUV} = 1.48$ a.u., for (b) the fit yields $\omega_{y,XUV} = 1.63$ a.u..

We take the same pulse as in Fig.2b, with time-dependent ellipticity, but we now add the time-dependent phase $\varphi(t) = 0.15(t - t_{0,XUV})$. This corresponds to the frequency shift for the y component. From the fits to the centers of the FROG-CRAB spectrograms taken separately for x and y components, see Fig. 2, we identify the two different carrier frequencies for the

x and y components. The frequency shift is accurately retrieved to be 0.15 a.u., i.e. about 4.1 eV.

We now measure the asymmetry. We align the IR polarization parallel to the major axis of the time-averaged ellipse and measure the spectra in the perpendicular direction, i.e., at 90 and 270 degrees, see Fig. 3a for $\cos^2 \gamma = 1$. The asymmetry in Fig. 3 (b) is computed for the case of no depolarization. It is markedly different from the case of time-independent phase φ , i.e. it is also sensitive to $\varphi(t)$. Crucially, the asymmetry is very sensitive to $\sin^2 \gamma$, see Fig. 3 (c,d) computed for $\sin^2 \gamma = 0.3$. Hence, the degree of coherence (or, alternatively, the depolarization) can be easily extracted by fitting the asymmetry obtained from the theoretical calculations for different values of $\sin^2 \gamma$ to the experimentally measured asymmetry. We also point out that one can take advantage of the great redundancy of this technique, by measuring the photo-electron signal along several angles and for several time delays. The

Figure 3: Photoelectron distributions at 90° and 270° and their asymmetry as a function of electron momentum. The IR field is polarized along the major axis of the XUV ellipse (along 0°) and the dashed lines indicate the region where we compute the asymmetry. (a,b) Perfectly polarized light, (c,d) partially polarized light ($\cos^2 \gamma = 0.7$). The asymmetry strongly depends on the degree of polarization.

Higher order terms (quadratic time dependence, etc.) are no different than first order terms; the chirp of the x and y components individually can be obtained by using FROG-CRAB, whilst the degree of coherence can be retrieved with the fitting procedure described above.

2. In addition, the authors only use a time-independent (de)polarization degree gamma, such that they cannot describe, e.g., a situation where first an unpolarized prepulse arrives, followed by a perfectly polarized main pulse.

Indeed, the degree of de-polarization we are using is approximately time-independent. Nevertheless, it does change in some of the examples considered, when the polarized and unpolarized components have somewhat different envelopes, see the answer to the question 4

below. The value of γ determines the overall relative weight of the two components but does not necessarily imply that their instantaneous contributions are time-independent.

We agree with the referee that considering strongly time-dependent de-polarization would be a logical next step, however, our technique is already able to characterize a vast majority of highly complex pulses, possibly more complex than those which will be present in real two-color HHG experiments, for example.

Our technique is the first, and for the moment the only one that can characterize time-dependent polarization state of chiral isolated attosecond pulses, solving the outstanding problem of distinguishing between the polarized and unpolarized components of such pulses. No other table-top technique that reconstructs other sources of light (harmonic XUV light [C. Chen *et al.* Science Advances **2** No.2 (2016)], femtosecond visible pulses [P. Carpegiani *et al.* Nature Photonics **11** 383-389 (2017)]) is able to reconstruct even the *time-independent* degree of (de-)polarization, let alone the time-dependent one. Extending our technique to time-dependent degree of de-polarization is left for future work.

We have modified the text accordingly, clearly stating that γ determines the overall contribution of the de-polarized component, while the instantaneous contribution may be time-dependent if the envelopes are different.

3. *It is not fully clear to me whether the assumption that the "unpolarized" contribution has equal x and y components is necessary or physically justified. According to my understanding, this term describes a randomly fluctuating contribution to the pulses, but I could just as well imagine a situation where there is such a contribution, but it is always linearly polarized in the same direction. E.g., this could occur in "standard" HHG with linear polarization of the driving pulses (such that all generated pulses are also linearly polarized), but where the IR pulse CEP is not well-stabilized, such that each attosecond pulse is slightly different.*

This is a very interesting comment, and we have performed additional calculations to address it.

First of all, let us note that there is a bit of terminological misunderstanding. The fact that the unpolarized light has equal contributions of x and y components is not an assumption: it is the definition of unpolarized light, the light with polarization changing randomly from shot to shot. There can be no excess along a specific direction.

That said, it is true that there could be random fluctuations along one specific direction and not the other. This is a different physical situation, which we have now analyzed. It means fluctuating ellipticity from shot to shot. We have simulated the case in which the y component fluctuated from shot to shot, so that the ellipticity fluctuated from 0.7 to 0.8. We incoherently summed the spectra computed for ellipticity of 0.7, 0.725, 0.75, 0.775 and 0.8 (see Fig. 4). From the measurement of the asymmetry, we retrieved the ellipticity of $\epsilon = 0.74$, which is essentially the average of the fluctuations (see Fig. 5).

This result has now been added to the supplementary material.

Figure 4: Angularly-resolved XUV+IR photoelectron spectra of averaged ellipticities (0.7, 0.725, 0.75, 0.775, 0.8) mimicking an experimental spectra with a fluctuation of the y component from pulse to pulse.

Figure 5: Fitting of the asymmetries A_1 and A_2 for several measurements of a pulse with a fluctuating ellipticity (from 0.7 to 0.8).

4. *The authors claim that it is basically trivial to obtain the x - and y -components of the pulse separately (“These spectrograms enable complete reconstruction of the XUV field components $F_x(t)$ and $F_y(t)$ individually.”) However, as far as I see, it is never mentioned what $F_x(t)$ and $F_y(t)$ actually are here. Since the unpolarized contribution has a random phase relation to the polarized part, the x -component and y -component of the full pulse also cannot be described by single functions $F_x(t) / F_y(t)$, but themselves consist of mixtures of “coherent” and “incoherent” radiation with a random phase between them. If the supposedly reconstructed $F_x(t)$ and $F_y(t)$ only corresponds to the coherent part, having these two functions already fully describes the “polarized” part of the pulse, and epsilon (and even a potential additional phase ψ as mentioned above) could be calculated from these functions trivially. In that case, it is not clear to me why an additional measurement would still be necessary. On the other hand, if the XUV field components mentioned in the text correspond to some mixture between the coherent and incoherent parts, it should be explained what is actually measured.*

This is an important point, which we hope we will clarify now.

The spectrogram measured by the full average signal (which is the experimental observable) is dependent on the distributions of both the polarized and unpolarized components. Hence, from two standard FROG-CRAB measurements (along x and y), one cannot disentangle the polarized and de-polarized components. The ellipticity will still be entangled with the degree of polarization.

The main point is that the unpolarized component cancels when we measure the asymmetry at two points which are separated by 180° , so that this observable contains information *only* on the polarized part. Combining both the average signal and the asymmetry measurements permits one to disentangle the polarized from the unpolarized (de-polarized) component.

We have tested this point in the manuscript. In the TDSE simulations we have performed, the envelope used for the unpolarized part was different from the envelopes used for the polarized components in both the x and y directions, i.e., $|\mathcal{F}_{unpol}|^2 \neq |\mathcal{F}_x|^2 \neq |\mathcal{F}_y|^2$. Of course, we have used the same envelope for the unpolarized component in the x and y directions, $|\mathcal{F}_{unpol,x}|^2 = |\mathcal{F}_{unpol,y}|^2 \equiv |\mathcal{F}_{unpol}|^2$, as one should (see previous question).

We then reconstructed those simulated pulses with our analytical model described in the manuscript. The accuracy of the reconstruction, for complicated time-dependent ellipticities and different degrees of polarization, validates the approximations we used in deriving the analytical expression.

The approximation we have used was as follows: In Eq.(27) of the Supplementary Information, we define $|\mathcal{F}_x|^2$ and $|\mathcal{F}_y|^2$, the x and y envelopes of the *polarized* component of the light (upon the Fourier transform). Then, in Eq.(28), when we write the expression for the unpolarized component, we approximate that the envelopes and their Fourier transforms for the x and y directions are similar to those of the polarized light, i.e., $|\mathcal{F}_{unpol,x}|^2 \approx |\mathcal{F}_x|^2$ and $|\mathcal{F}_{unpol,y}|^2 \approx |\mathcal{F}_y|^2$, and we neglect the corrections to this approximation, since the asymmetry is already relatively small and these would be higher-order corrections. The approximation is also justified on the grounds that the attosecond XUV light comes out of the high harmonic generation process, where it may not be perfectly polarized [Veyrinas *et al.*, Faraday Discussions **194** 161-183 (2016)] but should approximately share the same temporal envelope associated with the recollision process.

Even in the case where the above approximation is not valid (because of very low degree of polarization and/or very complicated time-dependent ellipticity), one can still use our technique without these approximations, but without relying on the analytical expression. The average signal given in Eq.(31) should be modified as

$$S(k_x, k_y) = g(k_0)^2 \left\{ k_{0x}^2 \left(\frac{|\mathcal{F}_{pol,x}|^2 \cos^2 \gamma}{1 + \epsilon(t_0)^2} + \frac{|\mathcal{F}_{unpol}|^2 \sin^2 \gamma}{2} \right) + k_{0y}^2 \left(\frac{|\mathcal{F}_{pol,y}|^2 \epsilon(t_0)^2 \cos^2 \gamma}{1 + \epsilon(t_0)^2} + \frac{|\mathcal{F}_{unpol}|^2 \sin^2 \gamma}{2} \right) \right\}, \quad (6)$$

where a new parameter $|\mathcal{F}_{unpol}|^2$ now needs to be obtained. With this formula, and one for the asymmetry without any approximations, one would extract all of these functions and parameters (\mathcal{F}_x , \mathcal{F}_y , \mathcal{F}_{unpol} , $\epsilon(t_0)$ and $\cos^2 \gamma$). Of course, now one has to perform global fitting of the TDSE calculations to the measurements, taking advantage of the redundancy of the data, measuring the signal for many electron emission angles and for many time delays.

5. *The isolated attosecond pulses are described by complex functions, and it is not mentioned that the real part should be taken to obtain the actual electric field (which must be real).*

The referee is correct. We have now corrected Eq.1 of the main manuscript.

6. *The authors talk of "the" isolated attosecond pulse, but then use a description that is clearly meant to be understood statistically, i.e., as describing a large ensemble of similar pulses, where the "polarized" part is identical each time, while the "unpolarized" part has a random phase in each realization. However, this is never explicitly mentioned (although it is acknowledged in their reply to the previous referee reports). Taking their statements as written, "the attosecond pulse" actually has a perfectly well-defined polarization state that simply depends on phi and beta. It is only under*

the assumption that there exist many copies of the pulse with different random phases ϕ and β that one could talk of a "polarized" and "unpolarized" contribution. This should be made very clear, as it clearly also has lead to some confusion in the discussion with the referees.

The Referee is completely correct. We have modified the text accordingly, describing this scenario as one particular origin of de-polarization: "One physical scenario leading to the presence of unpolarized component could be the fluctuations in the relative phase or amplitude of x and y components of the attosecond pulse from shot to shot".

7. *Since the "unpolarized" part has a random phase $e(i\phi)$ between x and y components, it seems superfluous to include an additional prefactor " i " before the exponential (as this can be removed by shifting ϕ by $\pi/2$). The same applies to the polarized part, should the authors follow my suggestion above and extend the method to allow arbitrary phase relations between the x and y components there as well.*

The referee is correct. We have removed the factor i in the unpolarized component. For the polarized part, since in the main manuscript we make the derivation in the frame of reference where the x and y components are orthogonal (which, as we have shown before, can be done by rotating the frame of reference using the linear measurement), we have decided to keep it.

REVIEWERS' COMMENTS:

Reviewer #2 (Remarks to the Author):

After careful reading of the authors' replies, re-revised manuscript and SI, I consider that the concerns I raised previously have been adequately addressed, namely with the study of the effect of CEP noise on the pulse reconstruction. Based on this, as well as on the replies to the other reviewers, I can recommend the re-revised manuscript and SI for publication in Nature Communications.

Reviewer #5 (Remarks to the Author):

The authors have addressed all of my comments to my satisfaction. The manuscript now presents a more complete picture and, in my opinion, the claims made about complete reconstruction are justified. I thus recommend publication.

Answer to Referee 2

After careful reading of the authors' replies, re-revised manuscript and SI, I consider that the concerns I raised previously have been adequately addressed, namely with the study of the effect of CEP noise on the pulse reconstruction. Based on this, as well as on the replies to the other reviewers, I can recommend the re-revised manuscript and SI for publication in Nature Communications.

We thank the referee for his/her positive feedback and we are glad to receive the recommendation to publish our manuscript in Nature Communications.

Answer to Referee 5

The authors have addressed all of my comments to my satisfaction. The manuscript now presents a more complete picture and, in my opinion, the claims made about complete reconstruction are justified. I thus recommend publication.

We thank the referee for valuing our work and we are pleased to receive recommendation for publication.